# Advanced Forecasting Modeling to Early Predict Powdery Mildew First Appearance in Different Vines Cultivars

Roberto Valori [1], Corrado Costa [2], Simone Figorilli [2], Luciano Ortenzi [2,3], Rossella Manganiello [2], Roberto Ciccoritti [4], Francesca Cecchini [5], Massimo Morassut [5], Noemi Bevilacqua [5], Giorgio Colatosti [6], Giovanni Pica [6], Daniele Cedroni [6] and Francesca Antonucci [2,*]

1   STAPHYT, Contract Research Organization (CRO) in the Fields of Agrosciences and Regulatory Affairs, Main Facility Italy, Via della Meccanica 28, 04011 Aprilia, Italy
2   Consiglio per la Ricerca in Agricoltura e L'analisi Dell'economia Agraria (CREA), Centro di Ricerca Ingegneria e Trasformazioni Agroalimentari, Via della Pascolare 16, 00015 Rome, Italy
3   Department of Agriculture and Forest Sciences (DAFNE), Tuscia University, Via S. Camillo De Lellis, s.n.c., 01100 Viterbo, Italy
4   Consiglio per la Ricerca in Agricoltura e L'analisi Dell'economia Agraria (CREA), Centro di Ricerca Olivicoltura, Frutticoltura e Agrumicoltura, Via di Fioranello 52, 00134 Rome, Italy
5   Consiglio per la Ricerca in Agricoltura e L'analisi Dell'economia Agraria (CREA), Centro di Ricerca Viticoltura ed Enologia, Via Cantina Sperimentale 1, 00049 Rome, Italy
6   Agenzia Regionale per lo Sviluppo e l'Innovazione dell'Agricoltura del Lazio (ARSIAL), Via Rodolfo Lanciani 38, 00162 Rome, Italy
*   Correspondence: francesca.antonucci@crea.gov.it; Tel.: +39-0690675213; Fax: +39-0690625591

**Abstract:** Eurasian grapevine is a widely cultivated horticultural plant worldwide, but it is more susceptible to powdery mildew. In recent years, the high cost and negative environmental impact of calendar-applied sulfur fungicides are leading research to find alternative remedies. In this study, the early prediction (three days) of the first appearance of powdery mildew infection, on two different Italian grapevine cultivars, was detected through a partial least squares discriminant analysis (PLSDA). The treatment indications of the "PLSDA" models (treatments according to the predictive model) were compared with those of the "Standard" (treatments according to the established agricultural practice of the area). This allowed the early containment of the disease, preventing its subsequent propagation. The model was built based on weather-climate data and phytopathological information collected on the "Untreated" control cultivar to monitor the natural spread of the disease (three years of training and two of tests). For both the cultivars and the two test years (2021 and 2022), the "PLSDA" models early predicted the first appearance of fungal disease, reducing the treatment number (about four) with respect to "Standard". In addition, analyses of key fruit quality parameters were conducted to evaluate the effectiveness of treatment reduction.

**Keywords:** grapevine; Trebbiano; Sangiovese; fungal infection; predictive statistical model; partial least squares discriminant analysis (PLSDA); qualitative analyses

## 1. Introduction

Vine is a profitable horticultural plant largely cultivated all over the world [1]. Powdery mildew, caused by *Erysiphe necator* Schw. (syn. *Uncinula necator*), is one of the fungal diseases that mostly affects the Eurasian species *Vitis vinifera* L. This species represents over 80% of the cultivated vines in the world [2]. Powdery mildew is an obligate biotrophic ascomycete which acts by infecting all the green tissues of the grapevine [3]. In addition, low levels of powdery mildew infection cause very substantial damage to both fruit and wine such as irregular ripening and deterioration of the berries, which can significantly affect the flavor [4]. All these result in an increase in acidity and a decrease in anthocyanins and sugars [5]. The development of powdery mildew takes place with a high relative humidity.

In fact, humidity, in the range of 40–100%, allows germination of the conidia and infection [6]. The host's best defense is to operate as soon as possible after the first appearance of infection. As reported by Stummer et al. [7], grape berries remain susceptible to powdery mildew until fruit set. Cortesi et al. [8] proved that primary infections, due to overwintering mycelium, being present in the vineyard in the early stages of vegetative awakening can be very dangerous and is the precursor of early and serious epidemics, which could lead to partial or total loss of the final crop. In these cases, it is necessary to make early use of fungicides, which can mitigate the role of the mycelium as a wintering structure.

Nowadays, one of the most widely used grapevine disease control strategies of powdery mildew involves the use of sulfur-based pesticides and fungicides [9]. Among the other fungicides most commonly used in vineyards management are those whose targets are enzymes involved in electron transport within the mitochondrial membrane and sterol biosynthesis [10]. Others, such as the methyl benzimidazole carbamates, include broad-spectrum systemic fungicides used against different groups of pathogens, with action even post infection, used mostly for foliar diseases and characterized by low utilization rates [11]. However, this has a great number of disadvantages, such as their negative impact on the environment and the high cost of chemicals. In addition, their use often carries the pathogens to rapidly develop resistance. For all these reasons, as reported by Redl et al. [12], there is a growing interest of modern viticulture in searching for new alternative environmental methods of protection. One of these involves the application of advanced algorithms for the development of specific forecasting models [13,14].

Generally, the containment of this disease is entrusted to preventive calendar applications of fungicides, which provide for sequential treatments using specific products only from the phase of differentiation of the bunches, the one considered most at risk. It is therefore difficult to defend crops using deterministic models based exclusively on the release of ascospores linked to climatic conditions. For all these reasons, such an early appearance of the fungus in the crop leads to a significant reduction in production, even when stricter defense calendars are applied [15].

The aim of this work is to build a multivariate statistical model based on partial least squares discriminant analysis (PLSDA) for the early prediction of the first appearance of powdery mildew infection. The model is based on a two-year training (2019, 2021) and a one-year test (2022) of sampling, all of which took place on the same vineyard and on two cultivars: Sangiovese and Trebbiano. This preliminary model is built on climatic data, collected via a weather station installed in the experimental vineyard, and on phytopathological information collected on the not treated control cultivar to monitor the natural spread of the disease. The result is the prediction, three days in advance, of the appearance of the primary fungal infection. This could allow early containment of the disease and its consequent propagation. In addition, to evaluate the effectiveness of sulfur-based treatment reduction, the main qualitative parameters (i.e., soluble solid content, pH and titratable acidity) of the berry were analyzed.

## 2. Materials and Methods

### 2.1. Experimental Design and Plant Materials

Three years (2019, 2021 and 2022) of field trials were carried out in the experimental vineyard of CREA (Research Centre of Viticulture and Enology) conducted by the Regional Agency for Development and Innovation of Agriculture of Lazio (ARSIAL) and positioned in Velletri (Central Italy, Rome; 41°40′12″ N latitude, 12°46′48″ E longitude) at 332 m above sea level. The grapevines analyzed belonged to two cultivars of the *Vitis vinifera*: Trebbiano Toscano (white grape) and Sangiovese (red grape). For more information on both the experimental plant and its agronomic management see Cecchini et al. [16].

Three different fungicide application strategies were compared: (i) treatments according to the conventional agricultural practice of the area (Standard) basing on the experts evaluations and referring to pre-established calendars of fungicide applications; (ii) treatments according to the predictive model PLSDA (PLSDA), which are carried out only if

the model gives the alert of the fungal first appearance; (iii) untreated control where no fungicides against powdery mildew were applied (Untreated). The treatments, established via the modelling of the PLSDA application, were different from the standard treatments only in terms of the time of anti-powdery mildew application and not the type of product used. A sulfur-based fungicide (i.e., Prosper® spiroxamine) was applied on both cultivars, with a knapsack sprayer using compressed air as propellant following the same procedure of Cecchini et al. [16].

The vineyard was carefully inspected once a week (for 6 weeks), to detect the incidence/frequency of powdery mildew lesions. This is equal to the total number of diseased bunches on a total of 50 bunches assessed per replicate for all the different classes (Standard, PLSDA and Untreated) for both the cultivars (Figure 1).

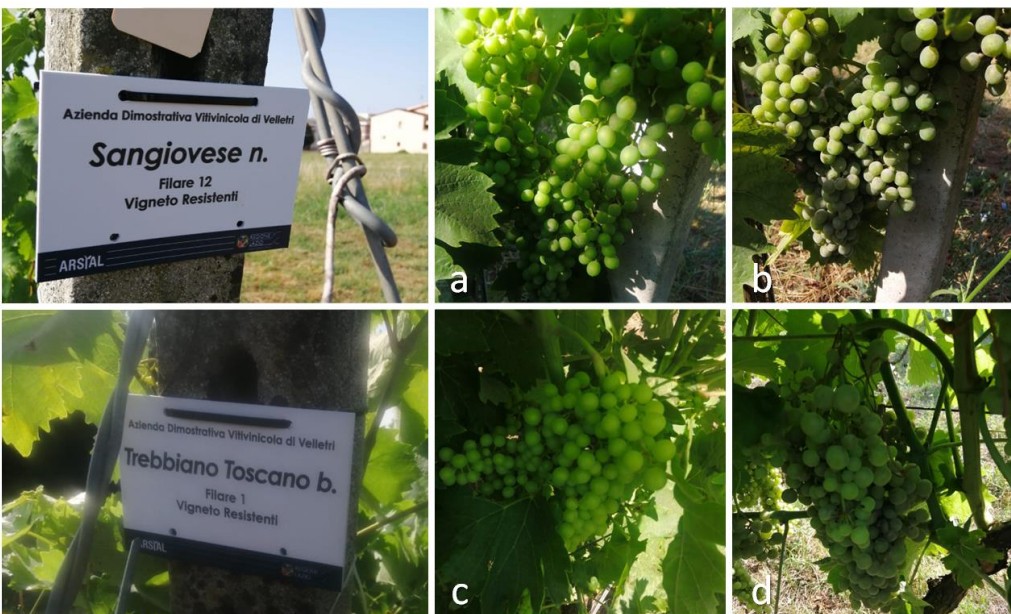

**Figure 1.** Grapevine samples of the two different cultivars (i.e., Sangiovese and Trebbiano) for both the diseased situations (i.e., (**a**,**c**) sane; (**b**,**d**) diseased).

Finally, the phenological stages were observed and defined in accordance with the international standard code "*Biologische Bundesanstalt, Bundessortenamt and CHemical industry*" (BBCH) [17]. The inspection was from setting to the beginning of ripening, 71 and 81 of the BBCH scale, respectively.

### 2.2. Climatic Data

Climatic data were taken from the Agenzia Regionale per lo Sviluppo e l'Innovazione dell'Agricoltura del Lazio (ARSIAL) weather station (RM10SPE, Cantina Sperimentale) located at the experimental vineyard CREA of Velletri (Rome) and collected during the crop years 2019, 2021 and 2022. Table 1 reports all the climatic variables collected in the three years of field trials: total hourly and daily leaf wetness (min), daily evapotranspiration (mm), total hourly and daily precipitation (mm), total hourly solar radiation (kJ/m$^2$), daily thermal sum at 0° (°C), daily thermal sum at 10° (°C), minimum, maximum and mean hourly air temperature at 2 m (°C), minimum, maximum and mean daily air temperature at 2 m (°C), mean daily ground temperature at −30 cm (°C), mean hourly ground temperature at −30 cm (°C), minimum, maximum and mean hourly relative air humidity at 2 m (%), minimum, maximum and mean daily relative air humidity at 2 m (%), mean hourly wind speed at 10 m (m/s) and mean daily wind speed at 10 m (m/s). All data (2019, 2021 and 2022) were used to explore the relation between climatic conditions and fungal disease. Meanwhile, to construct the models only the climatic data of 2021 and 2022 were used.

**Table 1.** Climatic variables collected in the three years of field trials by the ARSIAL weather station (RM10SPE, Cantina Sperimentale) located at the experimental vineyard CREA of Velletri (Rome).

| N. | Climatic Variables | Unit |
|----|-------------------|------|
| 1 | Total hourly leaf wetness | min |
| 2 | Total daily leaf wetness | min |
| 3 | Daily evapotranspiration | mm |
| 4 | Total hourly precipitation | mm |
| 5 | Total daily precipitation | mm |
| 6 | Total hourly solar radiation | $kJ/m^2$ |
| 7 | Daily thermal sum at 0° | °C |
| 8 | Daily thermal sum at 10° | °C |
| 9 | Minimum hourly air temperature at 2 m | °C |
| 10 | Maximum hourly air temperature at 2 m | °C |
| 11 | Mean hourly air temperature at 2 m | °C |
| 12 | Minimum daily air temperature at 2 m | °C |
| 13 | Maximum daily air temperature at 2 m | °C |
| 14 | Mean daily air temperature at 2 m | °C |
| 15 | Mean daily ground temperature at −30 cm | °C |
| 16 | Mean hourly ground temperature at −30 cm | °C |
| 17 | Minimum hourly relative humidity at 2 m | % |
| 18 | Maximum hourly relative air humidity at 2 m | % |
| 19 | Mean hourly relative air humidity at 2 m | % |
| 20 | Minimum daily relative air humidity at 2 m | % |
| 21 | Maximum daily relative air humidity at 2 m | % |
| 22 | Mean daily relative air humidity at 2 m | % |
| 23 | Mean hourly wind speed at 10 m | m/s |
| 24 | Mean daily wind speed at 10 m | m/s |

As previously reported by other studies [14,18–20], total hourly leaf wetness, total daily precipitation, mean daily air temperature and mean daily relative air humidity are the most important meteorological parameters that influence pathogen development. For this instance, mean daily data air temperature, mean daily relative air humidity, total daily precipitation and total hour leaf wetness, of the 3 years (2019, 2021 and 2022) of field trials (Day of Year (DOY) = 91–193) are reported in Table 2. It is possible to observe that the mean daily air temperature ranged between 4.9 °C (DOY = 92 of 2022) and 28.9 °C (DOY = 185 of 2022). In April, the minimum mean daily air temperatures rarely dropped below 10 °C, and the crop year 2019 showed the highest daily air temperature means value followed by the crop years 2022 and 2021 (12.42, 12.20 and 10.99 °C, respectively). In general, May showed the highest mean daily air temperatures compared to April, ranging from 8.9 °C to 28.9 °C, registered respectively on DOY = 135 of 2019 and on DOY = 147 of 2022. June and July were the hottest among the investigated months. Generally, minimum air temperatures rarely dropped below 18 °C, and the maximum was always above 25.6 °C. Mean daily relative air humidity followed the same trend of air temperature: mean daily relative air humidity never dropped below 33.5% (DOY = 185 of 2022) or rose above 94.9% (DOY = 124 of 2019). The highest mean values were recorded in April and May of 2019, which were the most humid. Meanwhile, 2022 was the driest. The total daily precipitation distribution differed significantly from the long-term trend: in 2019, 2021 and 2022 growing seasons, rainfall was significantly higher and lower than long-term means (+16%, −40% and −60%, respectively). Moreover, the period from DOY 91 to 193 of 2019 was the rainiest (about 332 mm) compared to 2021 and 2022 (about 182 and 109 mm, respectively). In detail, May 2019 was the rainiest month (215 mm) in contrast to April 2021 and 2022 (82 and 68 mm, respectively). Differently from 2019 and 2022 the period from DOY 152 to 182 of 2021 showed a significantly high amount of rain (about 68 mm). Finally, the maximum monthly mean leaf wetness was collected in May 2019, equivalent to the maximum rainfall (332.6 mm), in contrast to the minimum in July 2019.

**Table 2.** Mean monthly climatic data of the three experimental years (2019, 2021 and 2022) for the mean daily data air temperature at 2 m, mean daily relative air humidity at 2 m, total daily precipitation and total hourly leaf wetness variables.

| | | Ordinal Date (DOY) | Mean Daily Air Temperature at 2 m (°C) | Mean Daily Relative Air Humidity at 2 m (%) | Total Daily Precipitation (mm) | Total Hourly Leaf Wetness (min) |
|---|---|---|---|---|---|---|
| **2019** | April | 91–120 | 12.42 | 73.27 | 93.20 | 603.58 |
| | May | 121–152 | 13.31 | 82.59 | 215.40 | 903.17 |
| | June | 153–182 | 23.80 | 61.53 | 3.80 | 245.88 |
| | July | 183–193 | 25.60 | 61.44 | 19.00 | 62.08 |
| **2021** | April | 91–120 | 10.99 | 73.72 | 81.40 | 719.17 |
| | May | 121–152 | 15.81 | 72.38 | 35.00 | 499.71 |
| | June | 153–182 | 22.09 | 64.91 | 65.80 | 245.42 |
| | July | 183–193 | 24.12 | 63.58 | 0.00 | 89.29 |
| **2022** | April | 91–120 | 12.20 | 67.73 | 67.40 | 462.06 |
| | May | 121–152 | 19.18 | 62.41 | 16.40 | 312.83 |
| | June | 153–182 | 24.67 | 51.89 | 8.40 | 113.96 |
| | July | 183–193 | 25.77 | 49.67 | 17.20 | 37.67 |

### 2.3. Predictive Model Description

To predict the optimal time for applying sulfur compounds against powdery mildew, a PLSDA was considered. The PLSDA is a partial least squares (PLS) regression analysis with categorical response variables (Y-block; replaced by a set of dummy variables). The modelling approach, one per cultivar, Sangiovese and Trebbiano, was described in Menesatti et al. [21] and consists of two different phases: (1) model calibration and (2) field testing.

In this study, a statistical model was used. Generally, as reported by Lessio and Alma [22], the statistical model describes a situation varying according to probabilistic (and non-deterministic) events. This is constructed on a finite set of random variables depending on time and on the values that are assumed in the past. The model was built on a historical series of climatic data and phytopathological information (2019 and 2021 years), collected from April to July, a period in which powdery mildew could appear. This does not refer to the pathogen biological cycle, (e.g., ascospores or perithecia, fungal spore germination, germ tubes growing and branching out on the leaf surface), as requested using a deterministic model.

The datasets were constructed on the disease incidence collected on the untreated control (Y-block), for both Sangiovese and Trebbiano, on the ordinal date (starting from 1st April to 31st July of the same year), 24 mean daily variables (Table 1), grapevine phenological stages according to the BBCH-identification grapevine keys and epidemiological characteristics of powdery mildew to establish the infection risk (X-block).

The quantitative response of Y was transformed into a binary response variable of daily disease presence (1) or absence (0). The prediction was considered and modelled for the first appearance of the disease (presence/absence). The models, for Sangiovese and Trebbiano, were separately developed using a procedure written in MATLAB (version 7.1 R14).

The statistical models were constructed assuming that the first appearance of the pathogen depends on the parameters acquired the 3 days before. This elapse between the pathogen level and the X-block shifted i days before. In addition, the possibility was considered that the event could be related to the variable of specific adjacent (n) days to established different variables' weights. The daily prediction value was expressed as the probability of significant daily presence of the disease, and the best model was chosen following the procedure of Menesatti et al. [21]. PLSDA models for both Sangiovese and Trebbiano were calibrated using the 2019 and 2021 yearly data of disease incidence and tested on the 2022 yearly data.

### 2.4. Barry Quality Evaluation

Titratable acidity (TA) and pH were determined on about 500 g of homogenized fresh fruits following the procedure of Ceccarelli et al. [23]. TA content was expressed as g of tartaric acid equivalent (TAE) 100 g$^{-1}$ fresh weight (FW), and soluble solids content (SSC) was obtained on fruit juice with a digital refractometer (Refracto 30PX, Mettler Toledo, Switzerland) and expressed as g 100 g$^{-1}$ FW (Brix degrees).

### 2.5. Statistical Analyses

All the statistical analyses described below were carried out averaging over three replicates of each cultivar for each treatment. The whole dataset was subjected to one way analysis of variance (Kruskal–Wallis test) followed by a post hoc Dunn's test and linear correlation analyses. Both statistical evaluations were carried out with the software PAST (version 2.17v) [24].

## 3. Results

### 3.1. Model 2021

The indications of the treatments of the "PLSDA" models (treatments according to the predictive model PLSDA) were compared with those of the "Standard" (treatments according to the established agricultural practice of the area). For the tests of 2021, the models were trained with 2019 datasets. Figure 2 shows the comparison of the presence/absence of the first appearance of powdery mildew for the Trebbiano cultivar between the "Observed" and the "PLSDA predicted" infection for the year 2021. "Observed" refers to the untreated control plants where no fungicides against powdery mildew were applied.

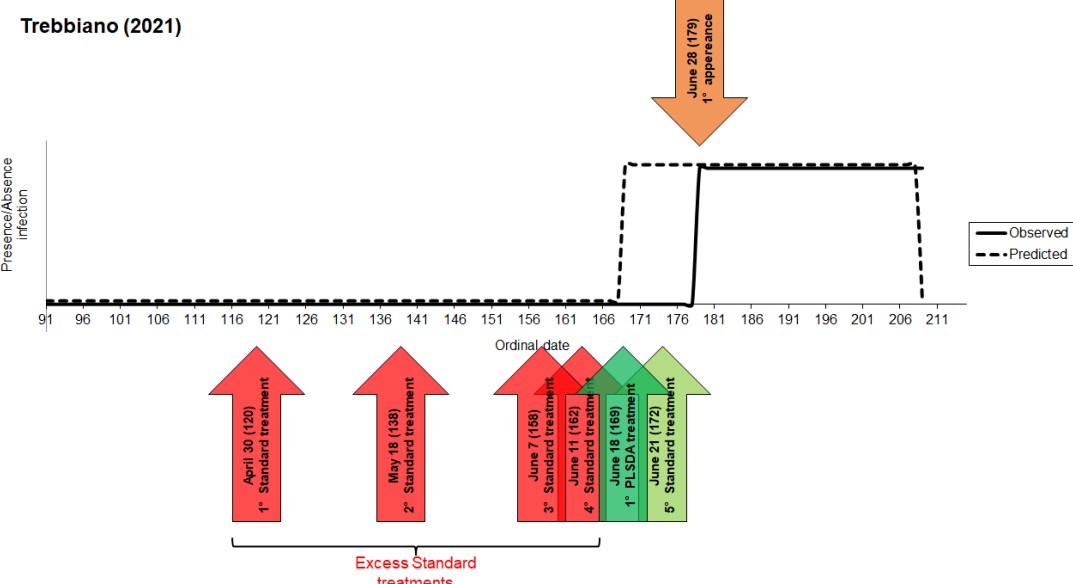

**Figure 2.** Plot of the presence/absence of the observed (continuous line) and predicted (dotted line) infections for the model trained on 2019 and tested on 2021 data for the cultivar Trebbiano. Only the central infection period (from 1 April (DOY = 91) to 31 July (DOY = 212)) is reported. The red arrows indicate the excess "Standard" treatment indications (treatments according to the established agricultural practice of the area); the light green arrow indicates the right "Standard" treatment indication; the dark green arrow indicates the "PLSDA" treatment indication.

In addition, the treatment indications of Standard and PLSDA are reported. It is possible to observe that the treatment indication for the "Standard" theses started in April: 30 April (DOY = 120), 18 May (DOY = 138), 7 June (DOY = 158), 11 June (DOY = 162) and 21 June (DOY = 172). Of these 5 treatments, the first 4 were found to be excessive as the

first appearance of powdery mildew occurred on 28 June (DOY = 179). The treatment of 21 June is the only effective one since it occurred 7 days before the powdery mildew first appearance (28 June, DOY = 179). Usually, these fungicides have a coverage of 10–15 days.

On the other hand, the "PLSDA" treatment indication is only for the 18 June (DOY = 169), clearly in line with the first appearance of fungal disease (28 June; DOY = 179).

Figure 3 shows the comparison of the presence/absence of the first appearance of powdery mildew for the Sangiovese cultivar between the "Observed" and the "PLSDA predicted" infection for the year 2021. For the "Standard" theses of Sangiovese cultivar, the treatment indications were the same as those for Trebbiano: 30 April (DOY = 120), 18 May (DOY = 138), 7 June (DOY = 158), 11 June (DOY = 162) and 21 June (DOY = 172). Also in this case, of these 5 treatments, the first 4 were found to be excessive as the first appearance of powdery mildew occurred on 28 June (DOY = 179). The treatment of 21 June is the only effective since it occurred 7 days before the first appearance of powdery mildew (28 June, DOY = 179). Usually, these fungicides have a coverage of 10–15 days.

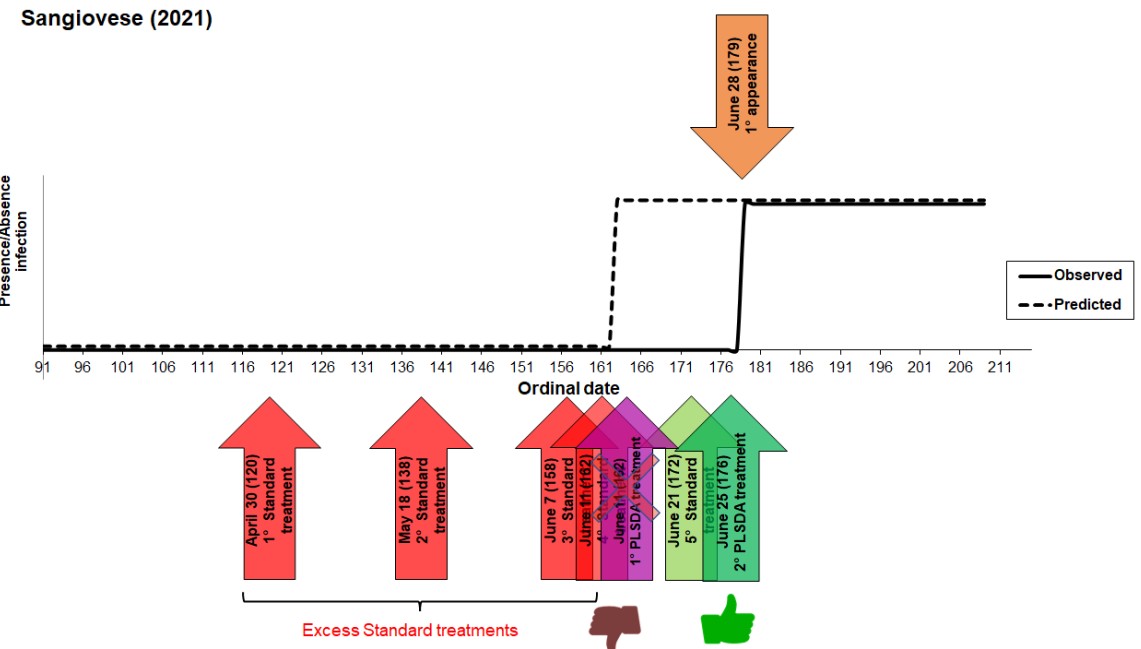

**Figure 3.** Plot of the presence/absence of the observed (continuous line) and predicted (dotted line) infections for the model trained on 2019 and tested on 2021 data for the cultivar Sangiovese. Only the central infection period (from 1 April (DOY = 91) to 31 July (DOY = 212)) is reported. The red arrows indicate the excess "Standard" treatment indications (treatments according to the established agricultural practice of the area); the light green arrow indicates the right "Standard" treatment indication; the purple arrow indicates the wrong "PLSDA" treatment indication; the dark green arrow indicates the right "PLSDA" treatment indication.

For the "PLSDA", one treatment indication, wrong, was given on 11 June (DOY = 162) and a second treatment indication (right) was given on 25 June (DOY = 176).

### 3.2. Model 2022

For the tests of 2022, the models were trained with 2019 and 2021 datasets. Figure 4 shows the comparison of the presence/absence of the first appearance of powdery mildew for the Trebbiano cultivar between the "Observed" and the "PLSDA predicted" infection for the year 2022. Also in this case, "Observed" refers to the untreated control plants where no fungicides against powdery mildew were applied.

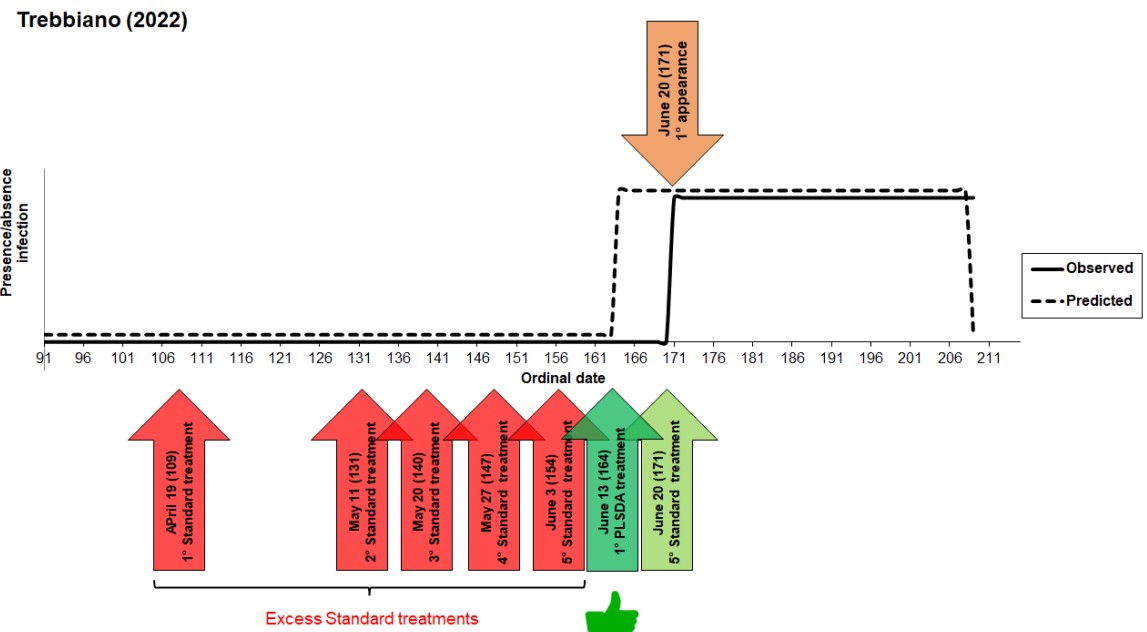

**Figure 4.** Plot of the presence/absence of the observed (continuous line) and predicted (dotted line) infections for the model trained on 2019 and 2021 data and tested on 2022 data for the cultivar Trebbiano. Only the central infection period (from 1 April (DOY = 91) to 31 July (DOY = 212)) is reported. The red arrows indicate the excess "Standard" treatment indications (treatments according to the established agricultural practice of the area); the light green arrow indicates the right "Standard" treatment indication; the dark green arrow indicates the "PLSDA" treatment indication.

It is possible to observe that the treatment indications for the "Standard" theses started in April: 19 April (DOY = 109), 11 May (DOY = 131), 27 May (DOY = 147), 3 June (DOY = 154) and 20 June (DOY = 171). Of these 6 treatments, the first 5 were found to be excessive as the first appearance of powdery mildew occurred on 20 June (DOY = 171).

For the "PLSDA", the treatment indication is only for the 13 June (DOY = 164), clearly in line with the first appearance of fungal disease (20 June; DOY = 171).

Figure 5 reports the comparison of the presence/absence of the first appearance of powdery mildew first for the Sangiovese cultivar between the "Observed" (carried out on untreated control where no fungicides against powdery mildew were applied) and the "PLSDA predicted" infection for the year 2022. Also in this case, the treatment indications for the "Standard" theses started in April: 19 April (DOY = 109), 11 May (DOY = 131), 20 May (DOY = 140), 27 May (DOY = 147) and 3 June (DOY = 154). Of these 5 treatments, the first 4 were found to be excessive as the first appearance of powdery mildew occurred on 13 June (DOY = 164).

For the "PLSDA", the treatment indication is only for the 9 June (DOY = 160), clearly in line with the first appearance of fungal disease (13 June; DOY = 164).

### 3.3. Powdery Mildew Disease Determination

The results of this study showed a significant and positive correlation between mean daily air temperature at 2 m and powdery mildew disease (r = 0.47 $p \leq 0.05$), while significant but negative correlation was found between relative humidity and mean daily air temperature at 2 m (r = $-0.34$ $p \leq 0.05$) and between total hourly leaf wetness and mean daily air temperature at 2 m (r = $-0.25$ $p \leq 0.05$). No significant correlation was found between mean hourly wind speed and infection (Figure 6).

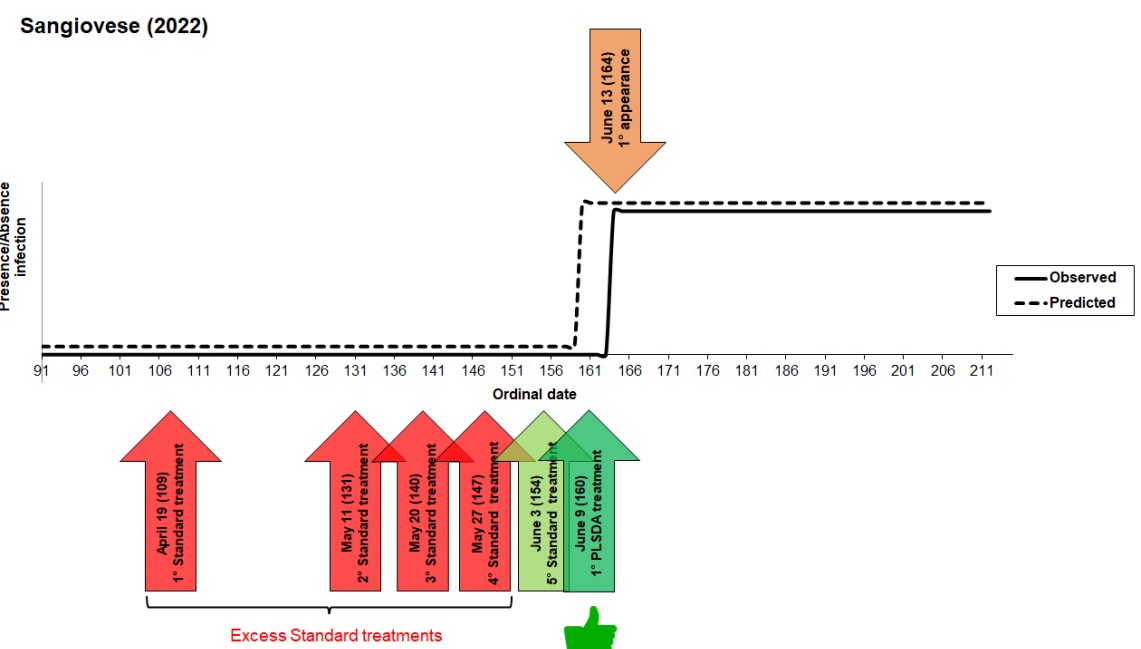

**Figure 5.** Plot of the presence/absence of the observed (continuous line) and predicted (dotted line) infections for the model trained on 2019 and 2021 data and tested on 2022 data for the cultivar Sangiovese. Only the central infection period (from 1 April (DOY = 91) to 31 July (DOY = 212)) is reported. The red arrows indicate the excess "Standard" treatment indications (treatments according to the established agricultural practice of the area); the light green arrow indicates the right "Standard" treatment indication; the dark green arrow indicates the "PLSDA" treatment indication.

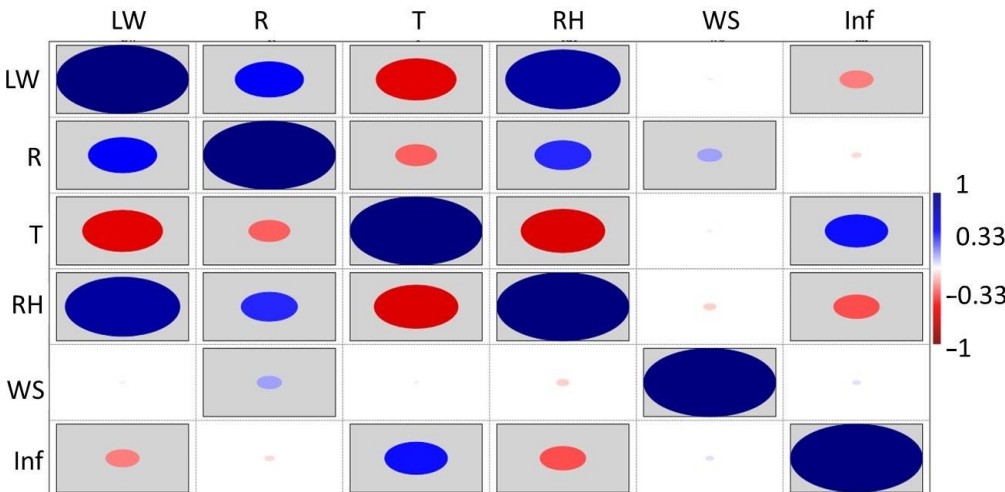

**Figure 6.** Pearson's correlation coefficients between climatic variables (i.e., mean daily air temperature at 2 m (°C) (T), mean daily relative air humidity at 2 m (RH), total daily precipitation (R), total hourly leaf wetness (LW) and mean hourly wind speed at 10 m (WS)) and powdery mildew disease (Inf.) for the three crop years (2019, 2021 and 2022).

During 2019, for both cultivars, disease was detected after DOY = 171, a few days after the rain occurred between DOY = 167 and DOY = 170. This rain allowed the first appearance of downy mildew infection, permitting the zoospores to penetrate the plant through the stomata, when the temperature, relative humidity and leaf wetness were optimal for disease development. The optimal air temperature and humidity conditions were when the daily mean temperatures were higher than 18 °C and the air humidity was higher than 60% (Figure 7).

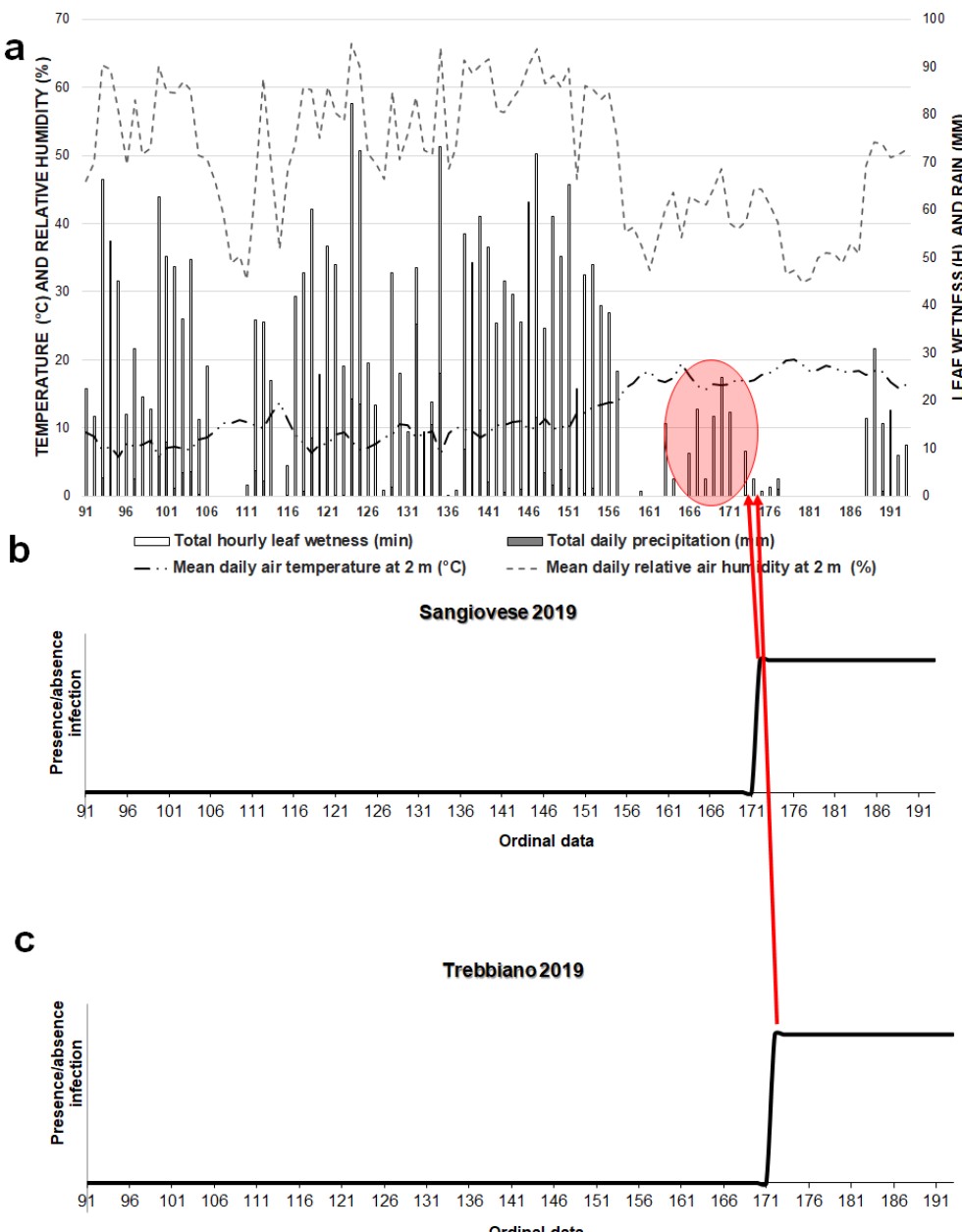

**Figure 7.** Climatic trend of: mean daily air temperature at 2 m (TEMPERATURE °C), mean daily relative air humidity at 2 m (RELATIVE HUMIDITY %), total hourly leaf wetness (LEAF WETNESS H) and total daily precipitation (RAIN MM) (**a**), in comparison with the plot of the presence/absence of the observed infections of powdery mildew for the year 2019 for both cultivars, i.e., Sangiovese (**b**) and Trebbiano (**c**). Red arrows indicate the first appearance of infection. The circle indicates rain causing the first infection.

Differently from 2019, during 2021, the infection symptoms appeared simultaneously for both cultivars at DOY = 175 (18 June) in untreated plants. Also for this year, the infection started after the rain occurred between DOY = 164 and DOY = 169, and after these rainy days, the optimum climatic condition for powdery mildew infection development was registered, characterized by mean temperatures higher than 18 °C and relative humidity of about 65% (Figure 8).

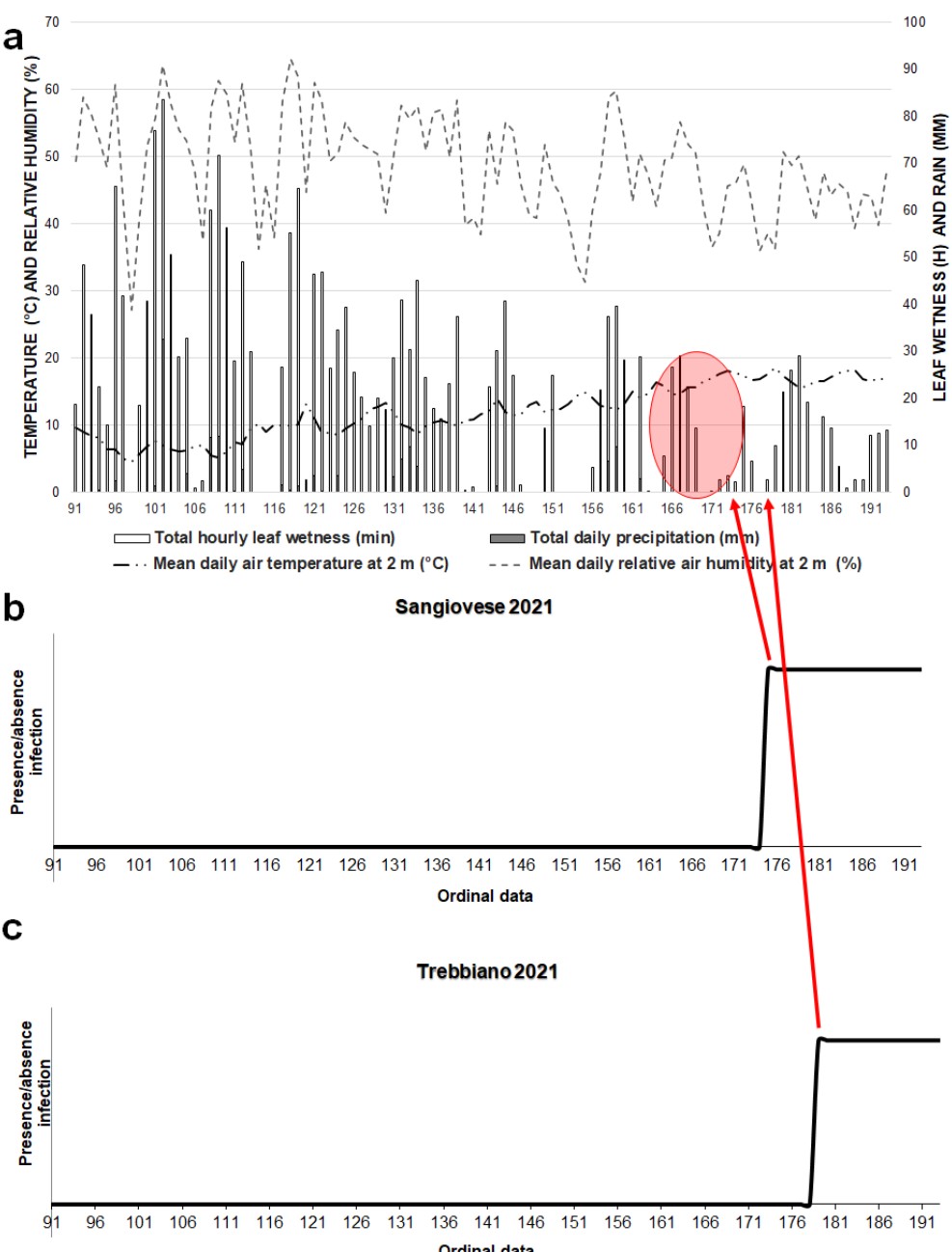

**Figure 8.** Climatic trend of: mean daily air temperature at 2 m (TEMPERATURE °C), mean daily relative air humidity at 2 m (RELATIVE HUMIDITY %), total hourly leaf wetness (LEAF WETNESS H) and total daily precipitation (RAIN MM) (**a**), in comparison with the plot of the presence/absence of the observed infections of powdery mildew for the year 2021 for both cultivars, i.e., Sangiovese (**b**) and Trebbiano (**c**). Red arrows indicate the first appearance of infection. The circle indicates rain causing the first infection.

A similar trend is observed for 2022. It is characterized by the anticipated comparison of the infection (DOY = 163 for Sangiovese and DOY = 170 for Trebbiano) in untreated plants with respect to 2021 (Figure 9). In detail, the spore dispersion occurred after the rain of DOY = 158 and DOY = 159, and the first appearance of disease occurred a few days after when the mean temperature increased to above 20 °C.

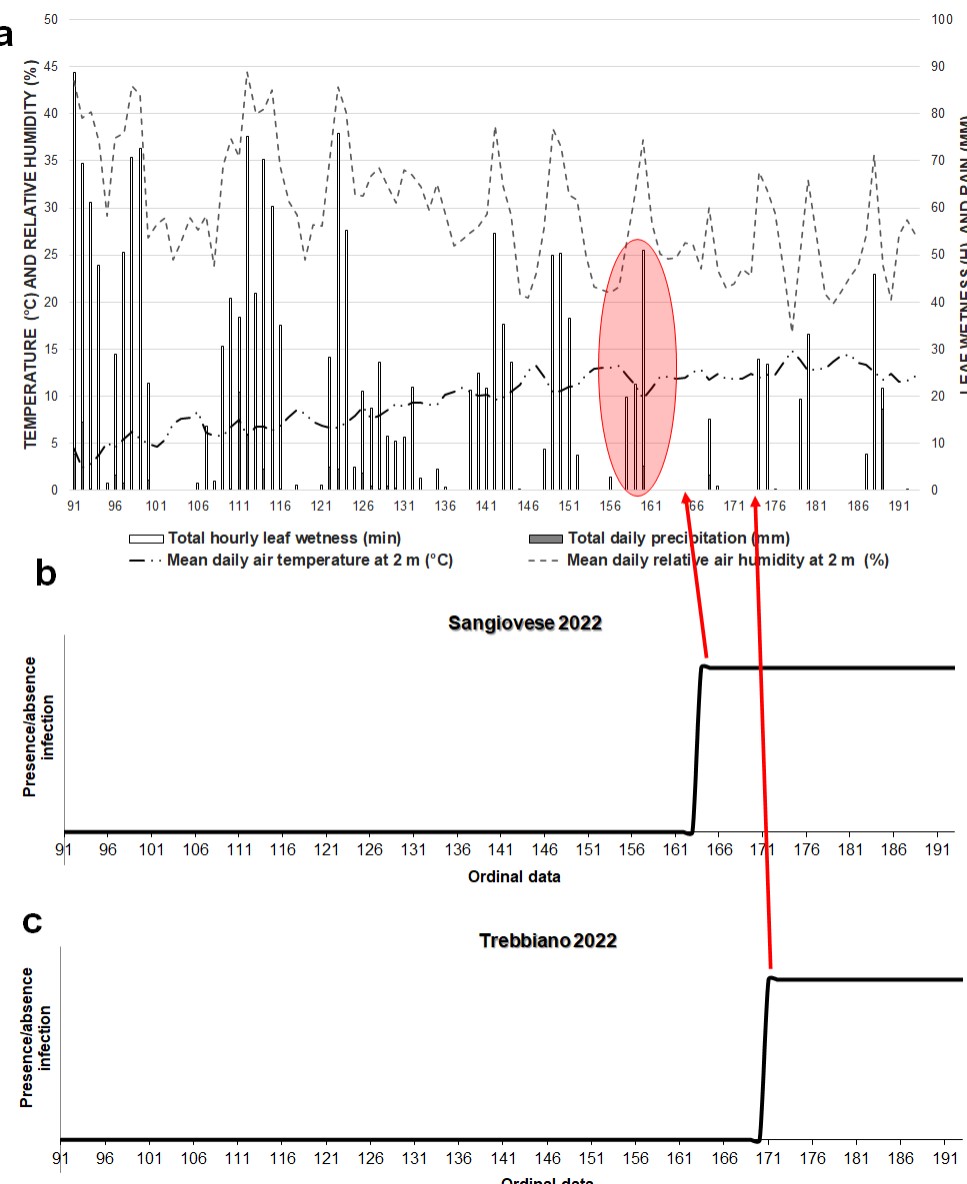

**Figure 9.** Climatic trend of: mean daily air temperature at 2 m (TEMPERATURE °C), mean daily relative air humidity at 2 m (RELATIVE HUMIDITY %), total hourly leaf wetness (LEAF WETNESS H) and total daily precipitation (RAIN MM) (**a**), in comparison with the plot of the presence/absence of the observed infections of powdery mildew for the year 2022 for both cultivars, i.e., Sangiovese (**b**) and Trebbiano (**c**). Red arrows indicate the first appearance of infection. The circle indicates rain causing the first infection.

### 3.4. Quality Assessment

The reduction of fungicides did not significantly affect the main qualitative parameters in both cultivars for the two different fungicide application strategies "PLSDA" and "Standard" (Figure 10). In detail, in Sangiovese, the solid soluble content (SSC) ranged from $22.1 \pm 0.2$ g 100 g$^{-1}$ FW for "PLSDA" plants to $22.0 \pm 0.3$ g 100 g$^{-1}$ FW for "Standard" ones. Similarly, for Trebbiano cultivar, there were significantly lower values: $20.1 \pm 0.8$ and $20.0 \pm 0.1$ g 100 g$^{-1}$ FW for "PLSDA" and "Standard" plants, respectively. No significant difference among the two different fungicide application strategies was also observed for pH and titratable acidity (TA) for both cultivars. In detail, Sangiovese pH ranged from 6.0 to 6.4 and TA ranged from $4.2 \pm 0.3$ g TAE 100 g$^{-1}$ FW to $4.5 \pm 0.4$ g TAE 100 g$^{-1}$ FW, respec-

tively lower and higher to that observed in Trebbiano (3.5 pH and $6.0 \pm 0.5$ g TAE 100 g$^{-1}$ FW for TA "PLSDA" and 3.2 pH and $6.4 \pm 0.3$ g TAE 100 g$^{-1}$ FW for TA "Standard").

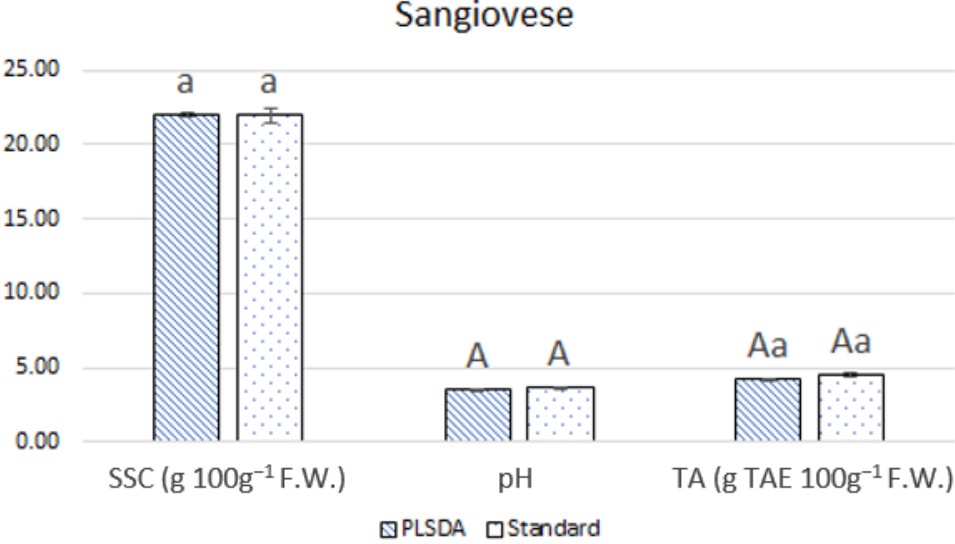

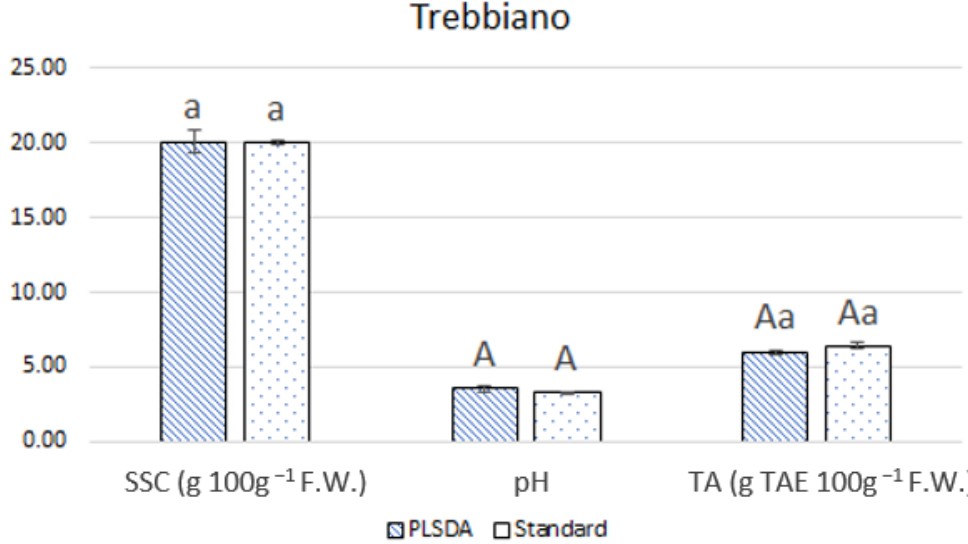

**Figure 10.** Qualitative parameters (i.e., solid soluble content (SSC), pH and titratable acidity (TA)) variability for the two different fungicide application strategies "PLSDA" and "Standard" for both the cultivars Sangiovese and Trebbiano, for the crop year 2022. Values belonging to the same traits without common letters are statistically different according to LSD ($p \leq 0.05$). Lower case letters are used for SSC evaluation, upper case letters are applied for pH comparison and lower case and upper case letters depict TA.

## 4. Discussion

The predictive statistical model developed in this study allowed the early prediction of the first appearance of powdery mildew infection on two different grapevine cultivars (i.e., Trebbiano and Sangiovese), thus enabling containment of the disease and its subsequent spread.

Generally, the grapevine has a high susceptibility to fungal attacks in the phenological phases of inflorescence beginning, flowering and fruit development, as demonstrated by the infection risk index calculated in the study by Fernández-González et al. [25], in

association with meteorological conditions that could facilitate the development of the infection. In addition, grapevine is very susceptible to infection during the early stages of fruit development, mainly in the first 1–2 weeks after fruit set and before veraison [26]. The fungal strain of powdery mildew is developing resistance to some commonly used fungicides. For this reason, as reported by Gaforio et al. [27], selecting less sensitive (e.g., the degree of susceptibility varies according to both the cultivar and environmental conditions) and high-quality cultivars represents an alternative management strategy of great importance. In this study, the degree of resistance indicated in the ampelographic descriptors of *Organisation Internationale de la Vigne et du Vin* (OIV) to powdery mildew in leaf and bunch (1 = very little, 3 = little, 5 = medium, 7 = high, 9 = very high) was 3 for Trebbiano cv and 2 for Sangiovese [28]. Berry infections occurring when fruit set are particularly damaging to quality, causing a significant reduction in yield and lower sugar concentration and higher acidity, also due to foliar infections affecting major losses of photosynthetic leaf area and thus delayed ripening [29]. Trebbiano is slightly more susceptible to powdery mildew than Sangiovese, as confirmed via qualitative analysis, in terms of SSC, pH and TA. A high susceptibility to powdery mildew in the grapevine is probably correlated to a low amount of phenolic compounds, especially flavonoid glycosides and hydroxy-cinnamic acid derivatives, and a high nitrogen intake in the leaf's constitution [30].

Therefore, powdery mildew affects grape berry development in morphological, physiological and biochemical properties of the final fruit [31]. Regarding biochemicals, powdery mildew alters water content, soluble/insoluble proteins, and fatty acid content, changing solid soluble content, pH and titratable acidity, which strongly affects the fruit qualitative properties [32,33].

The early prediction of the first appearance could lead to a reduction in the use of pesticides. This is because the growers use fungicides following calendar practices (i.e., whenever the treatment coverage ends) to manage powdery mildew in vineyards. From a sustainability assessment, limiting the use of plant protection products is beneficial not only for the environment, reduction of soil contamination and pesticide residues, but also for economic reasons. The reduction of treatment, as suggested by the models, did not significantly affect the main qualitative parameter investigated, confirming that a low number of treatments can efficiently control the development of fungal disease. The two investigated cultivars showed a similar trend for SSC, pH and TA ratio content values to those reported by many authors [34–36].

In the present study, the meteorological conditions occurring during the periods in which the highest powdery mildew concentrations were recorded generally corresponded to the optimal values for pathogen development and disease progression. In fact, precipitation, and humidity, at temperatures of 6–24 °C, favored the moistening of the cleistothecia, and therefore the dehiscence, facilitating the dispersion of the ascospores [19]. Relative humidity values between 53% and 97% and temperatures equal to or higher than 25 °C, such as those recorded in this study, are considered optimal for the development of infection [14,37]. In addition to the meteorological variables, the spatial distribution of the leaves could also influence the degree of powdery mildew infection in the grapevine: areas irradiated by the sun and with high temperatures inhibit the progress of infection, unlike those sheltered or in the shade where the disease develops [18].

This is a preliminary model, as only the first fungal infection appearance was monitored. Future monitoring will be necessary to make it repeatable, since the first appearance of powdery mildew is different according to different weather conditions; in addition, quantification of vinifiable and non-vinifiable grape yields and study of metabolites are needed.

## 5. Conclusions

This study represented a first step in model validation to overcome the typical limitations of the actual forecasting models for infections with the aim of rationalizing treatments and carrying out more timely pest control in a sustainability scenario. The results showed

that the first powdery mildew infection appeared when the temperature was higher than 18 °C and the relative humidity was higher than 55%. In addition, an important role was also played by the rainfall, which directly increased leaf wetness and relative humidity as well as facilitating the dispersion of the spores. Moreover, the results of the main qualitative parameters investigated highlight no significant difference of SSC and TA, confirming no negative effects of treatment reduction on berry global quality.

Finally, the developed models allowed identification and prediction of the times of real infection risk, to avoid the possible appearance of infection symptoms and possibly reduce fungicidal treatments for disease control with economic and environmental advantages. This information about risk infection alerts could be transferred to winegrowers, official regulatory organisms and the wine industry by means of websites or a cloud.

**Author Contributions:** Conceptualization, R.V., C.C. and F.A.; methodology, C.C. and F.A.; software, C.C., S.F., L.O. and F.A.; validation, R.V., C.C. and F.A.; formal analysis, C.C., R.C., G.P., G.C., D.C. and F.A.; investigation, R.V., C.C., F.C., M.M., N.B. and F.A.; resources, C.C.; data curation, C.C. and F.A.; writing—original draft preparation, R.M., R.C. and F.A.; writing—review and editing, R.M. and F.A.; visualization, C.C. and F.A.; supervision, F.A.; project administration, C.C. and F.A.; funding acquisition, C.C. All authors have read and agreed to the published version of the manuscript.

**Funding:** This article was funded by the Italian Ministry of Agriculture, Food and Forestry Policies (MiPAAF), sub-project 'Tecnologie digitali integrate per il rafforzamento sostenibile di produzioni e trasformazioni agroalimentari (AgroFiliere)' (AgriDigit national programme) (DM 36503.7305.2018 of 20/12/2018).

**Institutional Review Board Statement:** Not applicable.

**Informed Consent Statement:** Not applicable.

**Data Availability Statement:** Not applicable.

**Conflicts of Interest:** The authors declare no conflict of interest.

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
