# Peer review of "Advanced Forecasting Modeling to Early Predict Powdery Mildew First Appearance in Different Vines Cultivars"

_sustainability, doi:10.3390/su15032837_

Round 1
Reviewer 1 Report
Comments and two suggestions for the authors
The study presents a statistical model for prediction of appearance of grapevine powdery mildew infections. The aim with this work is to reduce the untargeted and unnecessary use of chemical fungicides. In my opinion this is an important study, as all research searching for possibilities to reduce the spread of man made chemicals into the environment, and fit well the scope for the journal.
The article is well written and I have only two suggestion for improving the manuscript: A) a picture of a infected vinegrape structure compared to a healthy crop would be nice for the reader. ) A short description of the most common fungicides and their impact on the environment may also interest the reader, (only 2-3 sentences inserted in the introduction part).
Author Response
Comments and two suggestions for the authors
The study presents a statistical model for prediction of appearance of grapevine powdery mildew infections. The aim with this work is to reduce the untargeted and unnecessary use of chemical fungicides. In my opinion this is an important study, as all research searching for possibilities to reduce the spread of man made chemicals into the environment, and fit well the scope for the journal.
The article is well written and I have only two suggestion for improving the manuscript:
- a picture of a infected vinegrape structure compared to a healthy crop would be nice for the reader.
- short description of the most common fungicides and their impact on the environment may also interest the reader, (only 2-3 sentences inserted in the introduction part).
Thanks for the suggestions and appreciations. As requested, we have included both the photos and some bibliographic references concerning the fungicides most used for grape powdery mildew.
Reviewer 2 Report
The topic of the manuscript is of particular interest to Eurasian vineyards. The undertaken research will contribute to better protection of the vines against fungal infections.
The set goal has been achieved. The purpose is set out quite extensively in the introduction of the manuscript.
The research material was properly described. Research methods correctly selected and developed.
The results of the obtained tests were statistically processed. They are presented in the form of charts and tables in a clear way.
Discussion of the results correct, concise summary.
In the opinion of the reviewer, the cited literature could be supplemented with even more up-to-date literature, especially pos. 14, 16.
After completion, I propose to accept it for printing.
Author Response
Comments and Suggestions for Authors
The topic of the manuscript is of particular interest to Eurasian vineyards. The undertaken research will contribute to better protection of the vines against fungal infections.
The set goal has been achieved. The purpose is set out quite extensively in the introduction of the manuscript.
The research material was properly described. Research methods correctly selected and developed.
The results of the obtained tests were statistically processed. They are presented in the form of charts and tables in a clear way.
Discussion of the results correct, concise summary.
In the opinion of the reviewer, the cited literature could be supplemented with even more up-to-date literature, especially pos. 14, 16.
After completion, I propose to accept it for printing.
Thank you for the suggestions and appreciations. As suggested, the literature has been integrated with even more recent references.
Reviewer 3 Report
The authors address an issue that is very emergent in the present moment, which is disease control strategy management in viticulture. This work has a very interesting premise and suggests models to predict the necessity of pesticide application to combat powdery mildew. However, there are certain important aspects that need revising and correction.
The writing should be revised throughout the manuscript to make it clearer and grammatically correct.
Why do the authors refer only to the months April to July? There are other months, before and after this period, that are vital for grapevine development and consequently disease manifestation.
I think it would be very important to consider for this study the OIV susceptibility descriptors for the chosen varieties. When discussing the appearance of symptoms in the two cultivars, this information should be integrated.
Specific suggestions:
Line 45: where it reads ascomycetes, it should read ascomycete
Line 60: Considering the sentence: “Nowadays, the only defenses for the containment of powdery mildew concern the use of sulfur-based pesticides and fungicides.” A reference should be added to this affirmation. Moreover, while the term “defenses for the containment” is not wrong in this context in my opinion it is more related to the plant’s immunological response. I would use “disease control strategies” instead
Lines 60-68: Several references should be added to this paragraph to support these affirmations. Moreover, I advise the authors to rewrite it in a more clear way.
Lines 60-74: I advise the authors to unite this paragraphs and to rewrite/remove some of the sentences of the second paragraph (lines 69-74) as it is quite repetitive
Line 86: The authors state that 2 cultivars are used in this study: Sangiovese and Trebbiano. The authors should also indicate the species, in this case Vitis vinifera and their degree of susceptibility to powdery mildew.
In Materials and Methods - Experimental Design and plant materials, the authors should explain why the varieties Sangiovese and Trebbiano were chosen for this study. Also in this section, the authors should explain with detail the pesticide treatments according to the standard model and the PLSDA module for each of the varieties (in case it is different). The paragraph from lines 119 to 122 should be rewritten as it is not clear. The authors should also explain in more detail how the disease incidence is calculated. It is not clear how the total, minimum and maximum number of affected branches are correlated top calculate the disease incidence.
In Line 126: The authors should explain the abbreviation BBCH and refer to the stages along the text
In Line 128: The authors should explain the abbreviation ARSIAL
Figures 1, 2, 3, 4 and 5 should be remade with different fonts as it is not possible to read the information on them.
In the result section, 2021 model, when the authors indicate that the first appearance if disease was in a certain date, do the authors refer to the control, “observed” infection? Please make it clear as you did for the 2022 model
Lines 307-311: This paragraph should be rewritten because it is not clear, specially the second negative correlation (between total hourly leaf wetness and what?).
Figures 6, 7 and 8: each of these figures should be divided in 3 parts – A, B and C, for instance – the graph containing the climate conditions, the graph indicating the appearance of disease in Sangiovese and the graph indicating the appearance of disease in Trebbiano. Moreover, the graph containing the climate conditions should be enlarged because the days are too close to each other in the ‘x’ axis and it becomes very difficult to read it and differentiate the ‘total hourly leaf wetness’ bars from the ‘total hourly precipitation’ ones. Moreover, in the x axis, it should be indicated that it refers to the number of days. Finally the legend for these figures should be rewritten as some parts are not clear (eg ‘The circle, the rain from which it came’).
Figure 9: Please explain in the legend of the figure what the letters in the graph over the bars mean
Lines 379 and 423: I advise the authors to refer to ‘grapevine’ instead of ‘vine’
Lines 384-388: After the sentence “Berry infections occurring when fruit set are particularly damaging to quality, causing a significant reduction in yield and lower sugar concentration and higher acidity, also due to foliar infections affecting major losses of photosynthetic leaf area, and thus delayed ripening.” The authors should add a reference
Line 388: When the authors state that “Trebbiano is slightly more susceptible to powdery mildew than Sangiovese”, this is a good place to include the OIV descriptors in the comparison
Lines 393-397: In this paragraph the authors write in the past tense which makes the test confusing. Are the authors referring alterations described in literature or in the present work? Please correct it.
Lines 398-405: This paragraph should be rewritten to make it easier to read. I suggest simpler, shorter sentences.
Line 406: I don’t understand why the authors say “However”. In my opinion while the result that is presented in this paragraph is interesting, it is no necessarily unexpected
Lines 411-413: The authors state that “Finally, as well-known, weather conditions (i.e., temperature, leaf humidity and rain) and leaf age are the main factors in the fungus development, while wind direction and speed determine the dispersion of spores [13].” In my opinion this paragraph should either be added to another paragraph or be deleted as it is because this information doesn’t add up to the discussion by itself.
In the last paragraph of the discussion it is important to also clearly add that these models should be rebuilt based on the data collected a larger set of years to give strength to the model. Moreover, while I agree that it would be beneficial to also analyze the metabolites, the authors should elaborate a bit more to explain why.
The first sentence of the conclusions is confusing as it is. The authors start by referring to “critical period” and end up indicating parameters. Please correct it and make it clear.
In my opinion, the authors should finish the conclusion with the main message from this work, which is currently in the middle of this section: “Developed models allowed identification and prediction of the times of real infection risk, to avoid the possible appearance of infection symptoms to the possibility to reduce fungicidal treatments for disease control with economics and environmental advantages. This information about risk infection alerts could be transferred to winegrowers, official regulatory organisms, and the wine industry by means of web sites or cloud.”
Author Response
Comments and Suggestions for Authors
The authors address an issue that is very emergent in the present moment, which is disease control strategy management in viticulture. This work has a very interesting premise and suggests models to predict the necessity of pesticide application to combat powdery mildew. However, there are certain important aspects that need revising and correction.
Thanks for the valuable corrections and suggestions. I hope the given answers are right.
The writing should be revised throughout the manuscript to make it clearer and grammatically correct.
The text has been deeply revised from a grammatical point of view as suggested.
Why do the authors refer only to the months April to July? There are other months, before and after this period, that are vital for grapevine development and consequently disease manifestation.
Thanks for the interesting question. In this study, a statistical model was used. Generally, the statistical model describes a situation varying according to probabilistic (and non-deterministic) events, such as, for example, all natural phenomena. In addition, a stochastic model is constructed on finite set of random variables depending on a parameter "t", which generally indicates time, and on the values that assuming in the past. The initialization of these random variables takes place through the identification of the probability distribution that characterizes each of these, through the statistical analysis of a database collected in the past. The database represents the probabilistic space of the values that the random variable can assume. In this specific study, the model is built on a historical series of climatic data and phytopathological information, collected from April to July, period in which powdery mildew could appear. In addition, April was considered because the budding phase begins for both cultivars. In this period may begin wintering of powdery mildew mycelium in buds. Given the extremely hot 2022 season, the harvest was brought forward by about 15 days to about mid-August. These are the mainly reasons because we refer to these months because we do not refer to the pathogen biological cycle, (e.g., ascospores or perithecia, fungal spores germination, germ tubes growing and branching out on the leaf surface), as requested by a deterministic model. In other words, the stochastic model builds a theoretical probabilistic prevision valid for the present, allowing to analyze the differences from the present conditions with respect to those predicted. This clarification has also been inserted into the paragraph “2.3. Predictive model descriptions”.
I think it would be very important to consider for this study the OIV susceptibility descriptors for the chosen varieties. When discussing the appearance of symptoms in the two cultivars, this information should be integrated.
Thank you for the important clarification. In the discussion section we added this information as requested: The fungal strain of powdery mildew is developing resistance to some commonly used fungicides. For this reason, as reported by Gaforio et al. [28], selecting less sensitive (e.g., the degree of susceptibility varies according to both the cultivar and environmental conditions) and high-quality cultivars, represents an alternative management strategy of great importance. In this study the degree of resistance indicated in the ampelographic descriptors of Organisation Internationale de la Vigne et du Vin (OIV) to powdery mildew in leaf and bunch (1 = very little, 3 = little, 5 = medium, 7 = high, 9 = very high) was 3 for Trebbiano cv and 2 for Sangiovese (both registered in the National Register of vine varieties) [29].
Specific suggestions:
Line 45: where it reads ascomycetes, it should read ascomycete
Done.
Line 60: Considering the sentence: “Nowadays, the only defenses for the containment of powdery mildew concern the use of sulfur-based pesticides and fungicides.” A reference should be added to this affirmation. Moreover, while the term “defenses for the containment” is not wrong in this context in my opinion it is more related to the plant’s immunological response. I would use “disease control strategies” instead
Done.
Lines 60-68: Several references should be added to this paragraph to support these affirmations. Moreover, I advise the authors to rewrite it in a more clear way.
Lines 60-74: I advise the authors to unite this paragraphs and to rewrite/remove some of the sentences of the second paragraph (lines 69-74) as it is quite repetitive
Thank you for your suggestion. We merged the two paragraphs, eliminated the redundant parts, and added new references.
Line 86: The authors state that 2 cultivars are used in this study: Sangiovese and Trebbiano. The authors should also indicate the species, in this case Vitis vinifera and their degree of susceptibility to powdery mildew.
We have integrated these important clarifications in the Discussion section as previously specified.
In Materials and Methods - Experimental Design and plant materials, the authors should explain why the varieties Sangiovese and Trebbiano were chosen for this study.
The cultivars were chosen because they are susceptible to the powdery mildew and because they are among the most widespread in Italy. Trebbiano and Sangiovese are, in fact, among the most used vines in the Italian wine industry.
Also in this section, the authors should explain with detail the pesticide treatments according to the standard model and the PLSDA module for each of the varieties (in case it is different). The paragraph from lines 119 to 122 should be rewritten as it is not clear.
We have modified the sentences and clarified the concept as requested.
The authors should also explain in more detail how the disease incidence is calculated. It is not clear how the total, minimum and maximum number of affected branches are correlated top calculate the disease incidence.
We apologize for the typo which we have corrected as and clarified as requested.
In Line 126: The authors should explain the abbreviation BBCH and refer to the stages along the text
We have explicated the acronymous and added the number of the BBCH scale as requested.
In Line 128: The authors should explain the abbreviation ARSIAL
Done
Figures 1, 2, 3, 4 and 5 should be remade with different fonts as it is not possible to read the information on them.
Done
In the result section, 2021 model, when the authors indicate that the first appearance if disease was in a certain date, do the authors refer to the control, “observed” infection? Please make it clear as you did for the 2022 model
Thank you for the suggestion. We have modified the sentences to better explain the differences between control, observed and predicted plants.
Lines 307-311: This paragraph should be rewritten because it is not clear, specially the second negative correlation (between total hourly leaf wetness and what?).
We have modified and rewritten the sentences as requested.
Figures 6, 7 and 8: each of these figures should be divided in 3 parts – A, B and C, for instance – the graph containing the climate conditions, the graph indicating the appearance of disease in Sangiovese and the graph indicating the appearance of disease in Trebbiano. Moreover, the graph containing the climate conditions should be enlarged because the days are too close to each other in the ‘x’ axis and it becomes very difficult to read it and differentiate the ‘total hourly leaf wetness’ bars from the ‘total hourly precipitation’ ones. Moreover, in the x axis, it should be indicated that it refers to the number of days. Finally the legend for these figures should be rewritten as some parts are not clear (eg ‘The circle, the rain from which it came’).
Thanks for your suggestions. We have modified both figures and captions as requested.
Figure 9: Please explain in the legend of the figure what the letters in the graph over the bars mean
Done
Lines 379 and 423: I advise the authors to refer to ‘grapevine’ instead of ‘vine’
Thanks for the suggested. We rewritten the indicated words.
Lines 384-388: After the sentence “Berry infections occurring when fruit set are particularly damaging to quality, causing a significant reduction in yield and lower sugar concentration and higher acidity, also due to foliar infections affecting major losses of photosynthetic leaf area, and thus delayed ripening.” The authors should add a reference
As suggested, a reference was added.
Line 388: When the authors state that “Trebbiano is slightly more susceptible to powdery mildew than Sangiovese”, this is a good place to include the OIV descriptors in the comparison
Done.
Lines 393-397: In this paragraph the authors write in the past tense which makes the test confusing. Are the authors referring alterations described in literature or in the present work? Please correct it.
We have corrected the sentences.
Lines 398-405: This paragraph should be rewritten to make it easier to read. I suggest simpler, shorter sentences.
We have better rewritten the paragraph.
Line 406: I don’t understand why the authors say “However”. In my opinion while the result that is presented in this paragraph is interesting, it is no necessarily unexpected
Done.
Lines 411-413: The authors state that “Finally, as well-known, weather conditions (i.e., temperature, leaf humidity and rain) and leaf age are the main factors in the fungus development, while wind direction and speed determine the dispersion of spores [13].” In my opinion this paragraph should either be added to another paragraph or be deleted as it is because this information doesn’t add up to the discussion by itself.
Thanks for the suggestion. We deleted the sentence as suggested.
The first sentence of the conclusions is confusing as it is. The authors start by referring to “critical period” and end up indicating parameters. Please correct it and make it clear.
In my opinion, the authors should finish the conclusion with the main message from this work, which is currently in the middle of this section: “Developed models allowed identification and prediction of the times of real infection risk, to avoid the possible appearance of infection symptoms to the possibility to reduce fungicidal treatments for disease control with economics and environmental advantages. This information about risk infection alerts could be transferred to winegrowers, official regulatory organisms, and the wine industry by means of web sites or cloud.”
Thanks for the useful suggestion that we accepted by modifying all the paragraph “Conclusions”.